# Dynamic causal modelling shows a prominent role of local inhibition in alpha power modulation in higher visual cortex

**Frederik Van de Steen**[1,2,3]*, **Dimitris Pinotsis**[4,5], **Wouter Devos**[1], **Nigel Colenbier**[6], **Iege Bassez**[1], **Karl Friston**[3], **Daniele Marinazzo**[1]

**1** Department of Data Analysis, Ghent University, Ghent, Belgium, **2** Vrije Universiteit Brussel, AIMS laboratory, Brussel, Belgium, **3** The Wellcome Trust Centre for Neuroimaging, University College London, London, United Kingdom, **4** Centre for Mathematical Neuroscience and Psychology and Department of Psychology, City—University of London, London, United Kingdom, **5** The Picower Institute for Learning & Memory and Department of Brain and Cognitive Sciences, Massachusetts Institute of Technology, Cambridge, Massachusetts, United States of America, **6** IRCCS San Camillo Hospital, Venice, Italy

* Frederik.van.de.steen@vub.be

**Data Availability Statement:** We used a publicly available EEG data set in this work and can be found here: https://physionet.org/content/

## Abstract

During resting-state EEG recordings, alpha activity is more prominent over the posterior cortex in eyes-closed (EC) conditions compared to eyes-open (EO). In this study, we characterized the difference in spectra between EO and EC conditions using dynamic causal modelling. Specifically, we investigated the role of intrinsic and extrinsic connectivity—within the visual cortex—in generating EC-EO alpha power differences over posterior electrodes. The primary visual cortex (V1) and the bilateral middle temporal visual areas (V5) were equipped with bidirectional extrinsic connections using a canonical microcircuit. The states of four intrinsically coupled subpopulations—within each occipital source—were also modelled. Using Bayesian model selection, we tested whether modulations of the intrinsic connections in V1, V5 or extrinsic connections (or a combination thereof) provided the best evidence for the data. In addition, using parametric empirical Bayes (PEB), we estimated group averages under the winning model. Bayesian model selection showed that the winning model contained both extrinsic connectivity modulations, as well as intrinsic connectivity modulations in all sources. The PEB analysis revealed increased extrinsic connectivity during EC. Overall, we found a reduction in the inhibitory intrinsic connections during EC. The results suggest that the intrinsic modulations in V5 played the most important role in producing EC-EO alpha differences, suggesting an intrinsic disinhibition in higher order visual cortex, during EC resting state.

## Author summary

One of the strongest signals that can be measured using EEG are so called alpha rhythms. These are neural oscillations that fall within the 8-12Hz frequency range. Alpha rhythms are most prominent when the eyes are closed and are seen at the electrodes placed at the back of the head. In this study, we studied the mechanism of alpha rhythms changes when

eegmmidb/1.0.0/. The code used for model fitting and plotting can be found on GitHub at: https://github.com/Frederikvdsteen/EO_OC_DCM.

**Funding:** This work is funded by the Ghent University Research Council (BOF17/GOA/004 to DM), UKRI (ES/T01279X/1 to KF and DP), funding for the Wellcome Centre for Human Neuroimaging (205103/Z/16/Z to KF) and the Research Foundation – Flanders (1267422N to FVDS). The funders had no role in study design, data collection and analysis, decision to publish, or preparation of the manuscript.

**Competing interests:** No Non-financial competing interests.

going from eyes-open to an eyes-closed state. We used a biologically plausible model including different neural populations. We focused on modelling connections within and between different neural sources of the visual cortex and how they are modulated when going from eyes-open to an eyes closed state. We found evidence that inhibitory neurons play an important role in alpha rhythms.

## Introduction

Alpha oscillatory EEG activity (i.e., 8-12Hz) during rest—or task-free recordings—is most pronounced during eyes-closed (EC) conditions, over the posterior cortex. During visual stimulation (i.e. eyes-open state, EO) alpha oscillations are suppressed (but see e.g. [1], who showed alpha suppression in a darkened room). Alpha-power modulations have also been observed during working memory [2, 3] and visual attention tasks (e.g. [4]). To date, the neural mechanisms underlying alpha activity and modulation (at the scalp level) remain an open question. Several studies have pointed to the role of the thalamus as driving source of cortical alpha [5–7]. More specifically, it is thought that the pulvinar and/or the lateral geniculate nucleus act as primary alpha-pacemaker(s). However, by using electrocorticographic recordings, a recent study showed that alpha waves in the cortex lead alpha activity in the thalamus [8]. Moreover, it has been suggested that cortico-cortical interactions play a prominent role—in addition to thalamo-cortical dynamics—in the generations of alpha rhythms [9] and that they are associated with conscious perception [10].

There are several difficulties in providing definitive explanations for alpha power differences between EO and EC. First, it is unclear whether we can recover signals from deep brain structures using non-invasive electrophysiological recordings, such as electro-and magnetoencephalography (EEG, MEG; [11, 12]). In order to study dynamics in terms of interacting brain regions from EEG and MEG signals, the so-called inverse problem needs to be solved (i.e. source reconstruction, [13, 14]). The accuracy of source localization solutions is to date still a matter of debate [11, 15, 16]. On the other hand, studies using intracranial recordings have high spatiotemporal resolution but due to the invasiveness, these studies are rather rare and usually involve a small number of electrodes and (clinical) sample size. Another disadvantage is that intracranial recordings do not cover the entire brain and are to some extent also susceptible to volume conduction. Finally, many electrophysiological studies that investigated the alpha-band—from a network perspective—have used measures such as coherence (i.e. modulus of the cross spectrum) and phase information to quantify (functional) connectivity. However, previous work has emphasized that the modulus (coherence) or arguments (phase) of the cross-spectra densities alone do not provide a unique or complete description of the underlying data generating process that produce spectral data features, such as spectral coherence [17]. These measures only provide a description of the statistical dependencies between observed signals but not how they are generated by (hidden) neural states.

One way to address some of these challenges is to use a forward (generative) model as in dynamic causal modelling (DCM; [18–20]). The generative model in DCM combines a biophysical and an observation model to describe the dynamics of hidden neural sources and how these neural states are translated into observed data. Furthermore, DCM provides a Bayesian framework to infer the unknown parameters of the model and to provide biophysical contraints on the neural dynamics. The biophysical model are sets of differential equations of coupled neural sources. Several experimental studies in human and non-humans and simulation studies have shown the face validity [18], construct validity [21], predictive validity [22] and

reliability [23] of DCM. Importantly, DCM for cross spectral densities was validated using an animal studies where DCM was able to correctly recover changes in synaptic physiology following neurochemical manipulation [24].

Here, we used DCM to model the underlying neural dynamics of observed spectral differences between EC and EO conditions, with a specific focus on alpha power. We employed DCM for cross spectral densities features, where both amplitude as well as phase information of the entire cross-spectra (including the autospectra) are used for inferring the underlying neural dynamics in terms of directed synaptic connections (i.e. effective connectivity, [17]). This means that phase information is also used in obtaining posterior estimates. We extended the current implementation by augmenting DCM with parameters characterizing state-dependent changes in intrinsic coupling [25, 26]. Inspired by a recent study [8], we modelled 3 distinct sources, assumed to be the main sources of EO-EC alpha power difference observed using EEG. These sources were the primary visual cortex (V1 collapsed across hemispheres, due to their proximity) and the bilateral middle temporal visual areas (V5), which were modelled using an established neural mass model based upon canonical microcircuits. Our main goal was to determine whether EO and EC alpha differences can be explained in terms of changes in either extrinsic connections (i.e. between sources) or changes in intrinsic connections (i.e. within a source) or their combination. We used parametric empirical bayes (PEB) to evaluate which specific connections show modulatory (i.e. condition-specific) effects [27, 28]. Finally, we examined the contribution of these modulatory parameters—to alpha power—in more detail, using a sensitivity analysis. We envision that the results here serve as a proof of principle that DCM can provide a mechanistic explanation of EO and EC differences in spectral activity. This is important since several studies have shown that the EO to EC alpha power difference is a neural marker of cognitive health [29–31].

## Materials and methods

### Data and pre-processing

In this study, 1-minute EEG recordings were taken from 109 subjects, during eyes open and eyes closed resting-state from the EEG Motor Movement/Imagery PhysioNet dataset [32, 33]. The data was acquired using the BCI2000 system http://www.bci2000.org). The EEG channels were placed on the scalp according to the international 10–10 system [34]. The data was provided in EDF+ format, containing 64 EEG channels, each sampled at 160 Hz. Eyes open resting-state was followed by the eyes-closed condition.

The data were pre-processed using EEGLAB running on MATLAB [35]. The 60Hz power line noise was first removed using the Cleanline EEGLAB plugin. Afterwards, the data were high-pass filtered using default settings, with a lower-cut-off of 1Hz. Then, a low-pass filter with high-cut-off of 45 Hz and default settings were applied. Periods of data contaminated with blink artefacts were repaired using independent component analysis. Bad channels were removed, based on visual inspection. Finally, the data were referenced to their average.

### Power spectral analysis

Our first goal was to confirm the well-known effect on posterior alpha power: During wakeful state, eyes closed are associated with much greater alpha power compared to having the eyes opened. Here, we estimated the power spectra from the last 10 seconds of the eyes-open period and the first 10 seconds of the eyes-closed condition. We choose not the use the full 1 minute resting-state recording because we had showed, in a previous study, that connectivity is non-stationary over 1 minute [36]. The power spectrum was obtained using Welch's method (i.e., pwelch.m command in MATLAB): The signal was divided into maximum 8 overlapping

windows with a 50% overlap between segments. Segments are obtained with a Hanning window and subsequently decomposed with discrete Fourier transform. This was repeated for every channel, subject and state (i.e., EO and EC). Permutation-based paired t-test was conducted by randomly assigning EO and EC labels to the entire power spectrum x channel data on a subject specific basis. We obtained 5000 permutations for the entire frequency x channel data space, during which the permuted t-values were retained. The t-value computed in each permutation was the same as the t-statistic that is used for a classical parametric paired t-test. The p-values were obtained by calculating the proportion of t-values obtained via permutation that exceeds the observed t-value, and the negative of the observed t-value. As such, a two-sided non-parametric paired t-test using permutation was performed. The major advantage of the permutation-based t-test is that it does not require any assumption about the distribution of the test statistic. The multiple comparisons problem was addressed using the Benjamini-Hochberg procedure for maintaining the false discovery rate (FDR) at 5% [37].

## Canonical microcircuit

In this study, brain sources are modelled with a neural mass model called the 'canonical microcircuitry' [38, 39].This model is equipped with four subpopulations per region: superficial and deep pyramidal cells (SP and DP), spiny stellate cells (SS) and inhibitory populations (II). Within each source, the subpopulations are coupled with so-called intrinsic connections, see **Fig 1** for a schematic presentation. The states in each subpopulation are described using the equations shown in **Fig 1**. Between source influences are mediated by extrinsic connections: Forward or backward (or both) connections, where forward connections originate from SP in one source and target SS and DP in another, while backward connections originate from DP and target SP and II. Exogenous (from other sources) inputs target SS. We used the spm_fx_cmc.m function that implements the equations described in Fig 1.

## Dynamic causal modelling for cross spectral data features

DCM is a Bayesian framework for inverting and comparing models of neural dynamics and the way these dynamics are translated into observations (in this case cross spectral data features). Therefore, it is useful to make a distinction between the neural model, which describes the hidden neural dynamics, and the observation model, which describes the mapping from neural states to observed responses. Usually, inference regarding the parameters of neural model is of interest (but see [40–44] for recent developments in multimodal fusion and applications of statistical decision theory in the context of DCM). A generative model is specified when the neural and forward model are combined and appropriately supplemented with prior constraints on the parameters. In this work, we used a specific DCM variant designed to deal with steady-state response called DCM for cross-spectral densities (CSD; [17, 45]). Here, the generative model specifies how neural dynamics—driven by endogenous fluctuations—map to observed cross spectral densities. By linearizing the model around its fixed point, the resulting transfer functions specify how the endogenous fluctuations are mapped, through neural dynamics and the forward model, to the observed CSD. The power spectrum of the endogenous fluctuations (innovations) is assumed to have a (parametrized) power law form: $g(\alpha, \beta, \omega) = \alpha\omega^{-\beta}$ with $\alpha$ and $\beta$ the parameters controlling the amplitude and the slope (or more precisely the rate of decay) of spectral densities of the innovations noise. These parameters are estimated for each region separately.

In order to infer condition dependent changes in intrinsic coupling, the current DCM implementation of the CMC model described above was supplemented with parameters

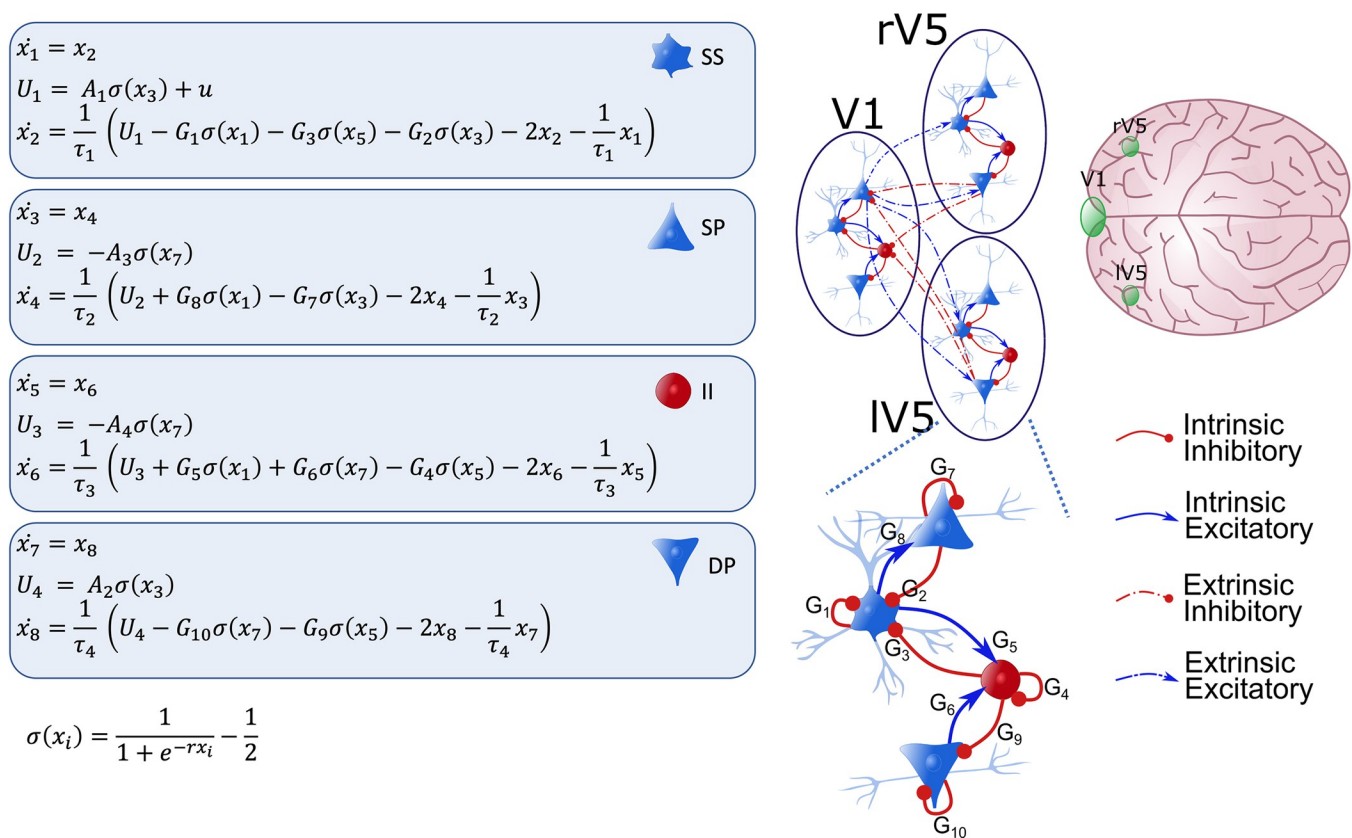

**Fig 1. Illustration of the Canonical Microcircuit Model (CMC) model.** Each source (V1, rV5 and lV5) comprises 4 neural subpopulations: spiny stellate cells (SS), superficial pyramidal cells (SP), deep pyramidal cells (DP) and inhibitory interneurons (II). Neural populations within a source are coupled with intrinsic connections (full arrows; bottom figure), while coupling between neural populations of different sources are extrinsic connections (dotted arrows). Red and blue arrows denote inhibitory and excitatory connections, respectively. The dynamics of the hidden (neuronal) states of each population can be described with the pairs of differential equations shown. There are four extrinsic connections: from SP to SS and DP (forward). Also, from DP to SP and II (backward). Intrinsic couplings are parametrized by $G_{1,\dots,10}$. Three regions comprise the network that is assumed to generate observed cross-spectral densities: V1 and left and right V5. These are shown on the top right. Forward connections were specified from V1 to V5 while backward connections were specified from V5 to V1. $\sigma(x_i)$ is a sigmoidal activation function which transforms post-synaptic potential into average spiking output, with $r$ a parameter controlling the steepness. Finally, the external input to a brain source is denoted with $u$ and enters SS. The free parameters shown in the figure are also listed in Table 1.

encoding these changes as following [25]:

$$G_i = G_i^A G_i^B$$

$$G_i^B = exp(X\theta_{Gi}^B)$$

Here, X encodes the conditions so that X = 0 for EO and X = 1 for EC condition. This implies that $G_i^A$ encodes baseline intrinsic connectivity and here corresponds to the EO-state. Consequently, $G_i^B$ encodes the modulatory gain of the $i$-th intrinsic connection associated with the EC-state. Connectivity and other parameters of the neural model are shown in Table 1. In other words, the baseline intrinsic connections are rescaled by a factor equal to $G_i^B$. Note that in Table 1, the parametrisation column indicates how the parameters in the right column are transformed into parameters that enter the neural state equations. This re-parametrization allows for sign constraints so, that for example, time constants are always positive. The parameters in the right column are those that effectively enter the model inversion scheme and here we used the '$\theta$' notation. For the re-parametrized parameters (left column) that enter the

**Table 1. Parameters of neural model (see Fig 1 for an illustration of the neural model).**

| | Description | Parametrization | Prior |
|---|---|---|---|
| $\tau_i$ | Postsynaptic time constant for subpopulation SS, SP, ii and DP | $(exp(\theta_k)[2\ 2\ 16\ 28])$ | $P(\theta_k) = N([0\ 0\ 0\ 0], 1/32)$ |
| $G^A_{1,..10}$ | Baseline intrinsic connectivity | $exp(\theta^A_G)$ $[1, 2, 4, 8]^*200$ | $P(\theta^A_{G1,..10}) = N(0, 1/8)$ |
| $G^B_{i = 1,...10}$ | Intrinsic connectivity modulation | $exp(\theta^A_G)$ | $P(\theta^B_G) = N(0, 1/4)$ |
| $A_{1,2,3,4}$ | Extrinsic connectivity | $exp(\theta_A)[1, \frac{1}{2}, 1, \frac{1}{2}]^*200$ | $P(\theta_A) = N(0, 1/16)$ |
| $B_{1,2,3,4}$ | Extrinsic connectivity modulation | $exp(\theta_B)$ | $P(\theta_B) = N(0, 1/8)$ |
| $\alpha, \beta$ | Amplitude and slope of the spectral innovations | $exp(\theta_{\alpha,\beta})$ | $P(\theta_{\alpha,\beta}) = N(0, 1/128)$ |

neural state and observation equations (see below), we used the notation that is also used in the SPM12 implementation. In the supplementary materials (S1 Appendix) we provide more information regarding the variational Bayesian inversion framework. Most of the parameters and their priors provided in Table 1 are the default priors of the DCM routine that was modified for implementing the intrinsic connectivity modulations (see spm_dcm_csd.m and spm_dcm_neural_prior.m and spm_fx_cmc.m of SPM12 version: 7487). The default priors were specified originally in [46] and were chosen to incorporate the known relative dissociation of the frequency content of superficial (high frequency) and deep pyramidal (low frequency) cells. Here only the prior on the intrinsic connectivity modulations were custom specified and set to a prior mean and variance of 0 and 0.25 respectively. This can be seen as a relatively uninformative prior which allows for the intrinsic connection to be modulated substantially during EC. Due to the re-parametrization, these priors are about log-scaling parameters and therefor the actual scaling ($G^B_i$) follows a Log-normal distribution (see supplementary materials S2 Appendix). The 5th and 95th percentile of this Log-normal distribution are .44 and 2.29 respectively. These bounds reflect halving and doubling baseline intrinsic connectivity. All the code for the analysis used in this paper are made publicly available at github (https://github.com/Frederikvdsteen/EO_OC_DCM).

The first 4 eigenmodes of the prior data covariance are used to project the channel data into a reduced sensor space:

$$y_{red} = Uy_{full}$$

Here U is the spatial projector that is obtained by taking the first 4 principal components of the prior data covariance (see [47] for more details). The cross spectral densities (CSD) that are used as data features are obtained from these 4 modes by fitting a Bayesian multivariate autoregressive model of order 12.

The forward model used here (the 'IMG'-option), treats each source as a patch on the cortical surface [48]. More specifically, the form used here is:

$$y_{full} = g(x, \theta_{obs})$$

$$g(x, \theta_{obs}) = \sum_{K=1:3} \Delta^K \sum_{j=1:8} \Psi_j x^k_j$$

$$\Delta^K = \sum_{n=1:6} \Theta^K_n Y^K$$

Here $y_{full}$ denotes the full EEG data and $g(x, \theta_{obs})$ that observation model. $\Delta^K$ is a Laplacian operator of region K, that is modelled as a mixture of $n = 1,..,6$ spatial basis functions. These

are obtained by taking the first 6 eigenvectors of the lead-field matrix of all the sources that fall within a sphere of 18mm of the cortical patch defined around the MNI coordinates of the 3 regions. These coordinates are: [0–88 4] for V1 and [−44 −68 0] and [42 −72 0] for left and right V5, respectively. Here the lead-field matrix was created using a default head model implemented in SPM12. $\Theta_n^K$ are the spatial parameters that are estimated during model inversion. The term $\sum_{j=1:8} \Psi_j x_j^k$ quantifies the contribution with weights $\Psi_j$ of the neuronal populations $x_j^k$ to the EEG data. The weights are free contribution parameters that are also estimated during inversion. Note that here only the depolarizing voltages of SP, DP and SS are allowed to contribute $y_{full}$ and are the same across the three regions. $\theta_{obs}$ denotes all the parameters of the observation model that are estimated during model inversion (i.e. the collection $\Psi_j, \Theta_n^K$). The DCM for CSD was inverted using a variational Bayesian inversion scheme described in [49] (see supplementary materials for more details).

DCMs with bad model fit (i.e. explained variance <50%) were removed from further analysis, in total 4 subjects were removed from the group-analysis. In S1 Fig, we provide the power spectrum of the 4 data modes of the removed subjects and 4 random subjects for comparison.

### Group level inference with parametric empirical Bayes

We used parametric empirical Bayes (PEB) to make inferences about extrinsic and intrinsic connectivity differences at the group level [27]. PEB uses a hierarchical model, which, at the first level, generates data from subject specific DCM parameters, and at the second level generates DCM parameters from group means, using a general linear model. The second level model characterizes between subject variability in terms of random effects. Here, we focused on group means of connectivity parameters. Using PEB, we obtained the posterior distribution of the (group mean) connectivity parameters and their condition-specific changes. In order to score the evidence for models with either extrinsic, V1 or V5 intrinsic connectivity modulations (or a combination thereof) we used Bayesian model reduction (BMR; [27]). We compared models with or without the following three sets of parameters: extrinsic modulations, V1 intrinsic modulations and V5 intrinsic modulations so that in total, 8 models were created. This included a null model with no connectivity modulations. We also used a greedy search algorithm and Bayesian model reduction to prune second-level parameters from the PEB model with lowest evidence until we obtained 256 'best' models. Bayesian model averaging (BMA) was subsequently applied to the reduced models to provide parameter estimates that accommodated for uncertainty over pruned or reduced models [50]. Inference of second level parameters—encoding group-mean intrinsic connectivity modulations—used the posterior probability (Pp), that quantifies whether a model including this parameter explains the data better than a model without. This has the advantage that the full parameter covariance matrix is used when comparing models, see [27] for details. The posterior means after BMA with Pp >.95 are treated as 'significant' in the sense that there is strong evidence for their contribution to the data.

## Results

### Power spectral results

Fig 2 displays the FDR-thresholded map of the frequency-by-channels t-values in image format. In addition, a 2-D topographical plot of the mean spectral power difference at 10.6Hz is shown and the mean power spectrum across subjects and 4 posterior electrodes (PO7, O1, O2 and PO6) are shown.

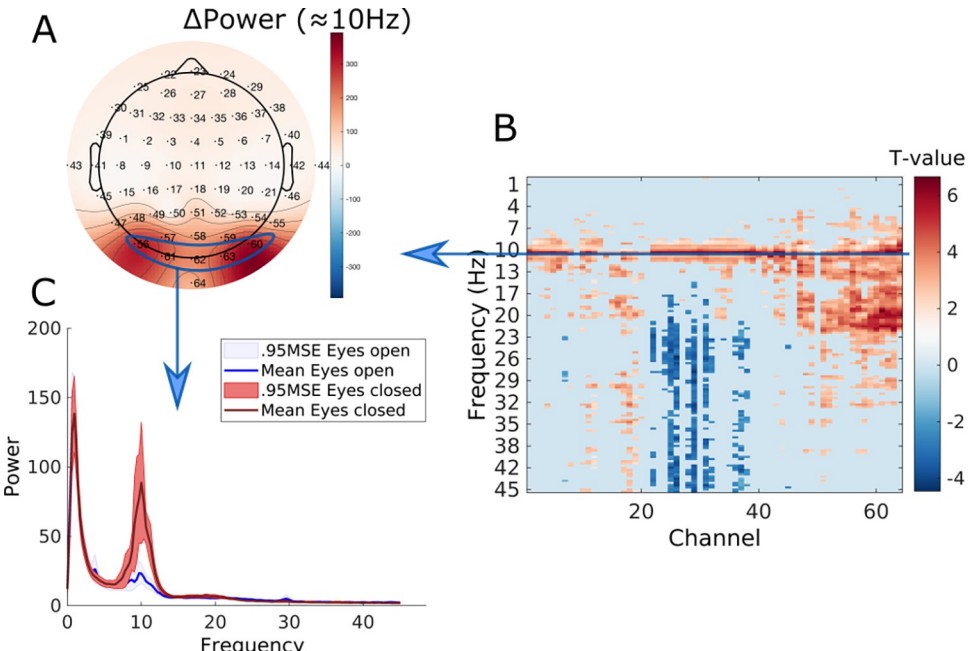

**Fig 2. Results of the power spectral analysis.** Panel A shows the topographic plot of the mean power difference (Δ Power; power = Db/frequency)) at 10.6 Hz. The FDR-thresholded t-values of all Channels x Frequencies are shown in image format in panel B. The mean power spectrum and 0.95 standard error intervals about the mean over channels PO7, Oz, O2 and PO8 for EO and EC are plotted in panel C.

The key things to note are significant differences around 10.6Hz. This difference was most pronounced over the posterior electrodes. However, a global effect can be observed in terms of statistical significance. Furthermore, positive and negative effects in higher frequency ranges were found. The positive effects were largely posteriorly localized (e.g., PO7, Oz, O2 and PO8; up to 23Hz), while the negative effects were localized to frontal electrodes (e.g., AF3, AF4, AF7, AF8; between 20 and 45Hz).

## PEB model selection and parameter averaging

As a first step, we created 8 alternative PEB models to test which connectivity modulations are related to the difference between EO to EC conditions. The 8 PEB models were created by taking the combinations of the following three parameter sets: with or without extrinsic modulations, with or without V1 intrinsic modulations and/or with or without V5 intrinsic connectivity modulations. This also includes a Null model with no extrinsic nor intrinsic connectivity modulations. In **Fig 3**, the log-evidence differences of the 8 models are shown.

These differences are with respect to the full PEB model. We found that the full model had the largest evidence. The second-best model contained both V1 and V5 intrinsic modulations. The log-evidence difference between the best and second-best model was 15.23. This is larger than 3, which can be considered as very strong evidence in favour of the full model. In short, we found that both extrinsic and intrinsic modulations in V1 and V5 play an important role in explaining differences between EO and EC conditions. However, an interesting pattern can be observed. We see that models without V5 intrinsic modulations had much lower evidence relative to models including V5 intrinsic modulations. This suggest that V5 intrinsic modulations were relatively important for explaining EO vs EC differences.

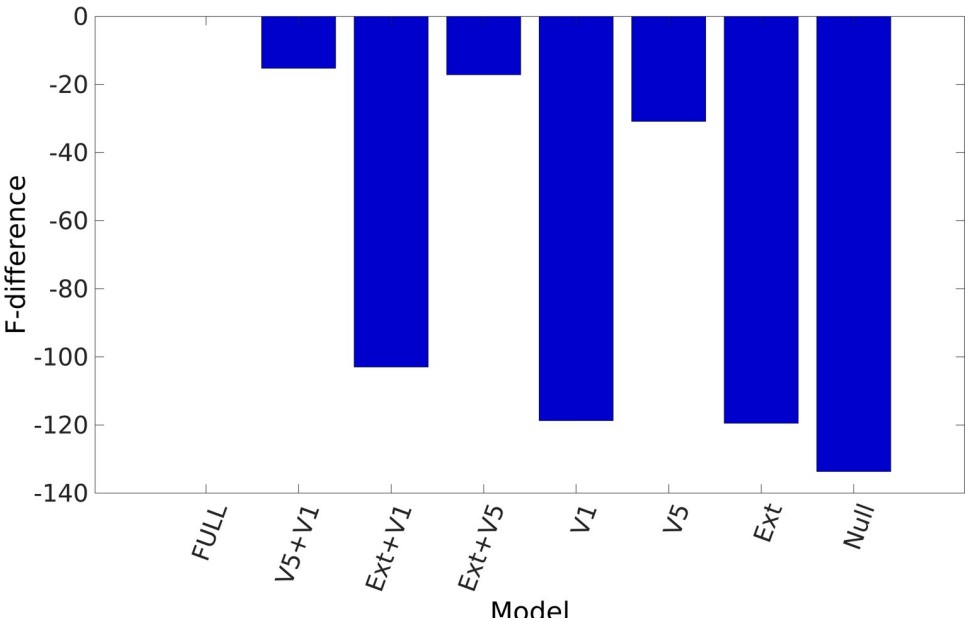

**Fig 3. PEB model selection.** The bar graph of the free energy (i.e., log evidence) differences from the full model (i.e. the model with extrinsic, V1 and V5 connectivity modulations) are shown for the 8 PEB models considered. The models were formed by creating combinations with or without the following three parameters sets: extrinsic modulations (Ext), V1 intrinsic modulations (V1) and V5 intrinsic modulations (V5). We observe that the full model has the highest (approximate) model evidence. In addition, models without V5 intrinsic modulation have smaller evidence compared to models that included V5 intrinsic modulation.

The Bayesian model average (BMA) estimates and 90% Bayesian confidence intervals of the four extrinsic modulations can be found in **Fig 4**. We observe that the forward extrinsic connections from V1.

The group analysis of the intrinsic connectivity modulation can be found in **Fig 5**. In V1, we observed a significant reduction of the inhibitory connection $G_3$ (II → SS) and $G_1$ (SS → SS). The excitatory connection $G_5$ (SS → II) increased during EC. The pattern for left and right V5 are similar: reduced modulation of the inhibitory connection of $G_3$ (II → SS) and $G_1$ (SS → SS). and increased inhibition of $G_9$ (II → DP).

In order to characterise the contributions of the intrinsic and extrinsic connectivity modulations on the power spectrum, we performed a sensitivity analysis. Briefly, for each modulation parameter, we examined the effect of a small parameter increment on the predicted power spectrum of the (reduced-) data of the posterior electrodes. More specifically, we added a small increment ($e^{-6}$) to the posterior mean of a certain parameter, while keeping the posterior means of the other parameters fixed. This was repeated for all the parameters of interest separately. In total, we performed the sensitivity analysis at the posterior means of 33 parameters. Technically, we are numerically evaluating the Jacobian of the generative model with respect to the extrinsic and intrinsic connectivity modulations at their posterior means ($\frac{\partial O}{\partial \theta}$, with $O$ the powerspectrum) over the different frequency bins. This was repeated for every subject separately and subsequently averaged over subjects. The results are reported in **Fig 6** together with the posterior mean of the group-BMA.

Simply put, positive (red) values indicate that an increase of a parameter at the posterior mean would result in increased power at that specific frequency bin. A negative value (in blue) means an increase of a particular parameter results in a decreased power. Changes in intrinsic connectivity have a larger effect on the power spectrum compared to extrinsic connectivity

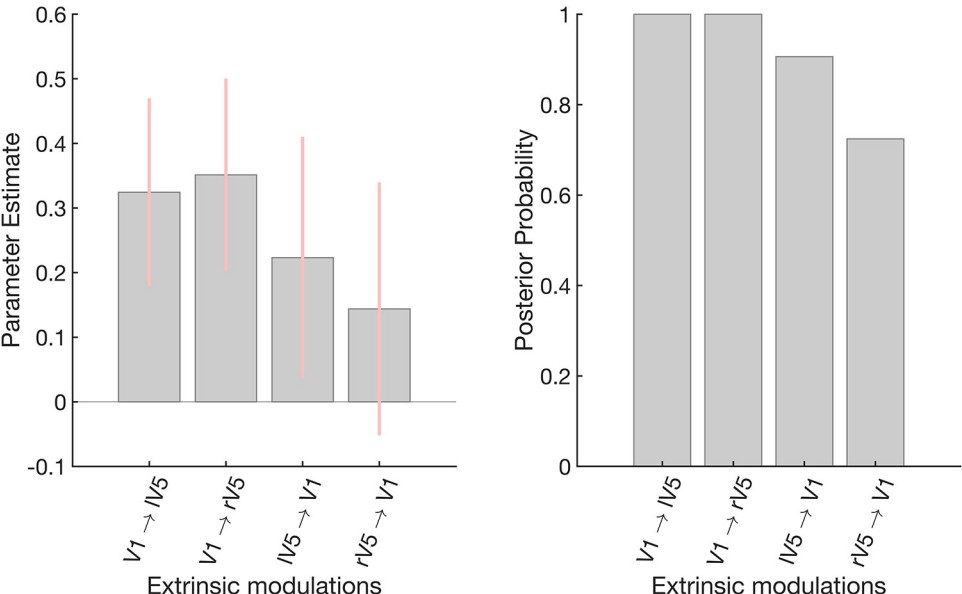

**Fig 4. Extrinsic connectivity modulations.** This figure shows the group-level results of the extrinsic connectivity modulations associated with EC states (relative to EO). More specifically, the mean posterior and 90% Bayesian confidence intervals (pink) after the greedy search algorithm and Bayesian model averaging are shown. Note that the estimated parameters are log-scale parameters (i.e., the parameters in the right column of Table 1 ($\theta_B$), but here we used different x-axis labelling for clarity.). The right panel shows the posterior probabilities of the extrinsic connections. increase during EC. The backwards connections appear to increase as well, however, the evidence is not conclusive (Pp<0.95). The posterior probabilities (Pp) are 100%, 100%, 90% and 72% for V1 → lV5, V1 → rV5, lV5 → V1 and rV5 → V1, respectively.

and this is most pronounced for V5. In addition, we see clearly that the sensitivities are most pronounced within the alpha band (i.e., around 10Hz). If we consider the significant intrinsic modulation, we observe largest negative sensitivities for the inhibitory $G_3$ (II → SS) and $G_1$ (SS → SS) modulations and positive sensitivities for the excitatory $G_9$(II → DP) modulations in left and right V5. The sensitivities of the significant modulations in V1 are much less pronounced compared to the sources in V5. Interestingly, the inhibitory $G_4$ (II → II) connection in V5 seems to have the largest sensitivities as well but are not significant parameters in the PEB analysis. This suggest that a reduction in $G_4$ in V5 could potentially be important for enhancing alpha power. Nevertheless, the current parameter configuration appears to suffice (in terms of the accuracy-complexity trade-off; see supplementary materials S1 Appendix) for fitting the observed responses.

In summary, we found evidence that both extrinsic modulations between V1 and V5—as well as intrinsic modulations within V1 and V5—play an important role in the genesis of EO-EC power spectral differences. In addition, we found that the intrinsic modulations in bilateral V5, in particular the inhibitory connections, seem to play the greatest role. This speaks to the importance of local [dis]inhibition, within higher order visual cortex.

## Discussion

In this work, we investigated the role of intrinsic and extrinsic connections within the occipital cortex in the generation of EO and EC alpha power differences. Using a publicly available data set, we first replicated previous findings that alpha-power is most pronounced during EC condition at posterior channels. Then, using DCM followed by PEB, our analysis showed that the model with the largest evidence contained both extrinsic and intrinsic connectivity

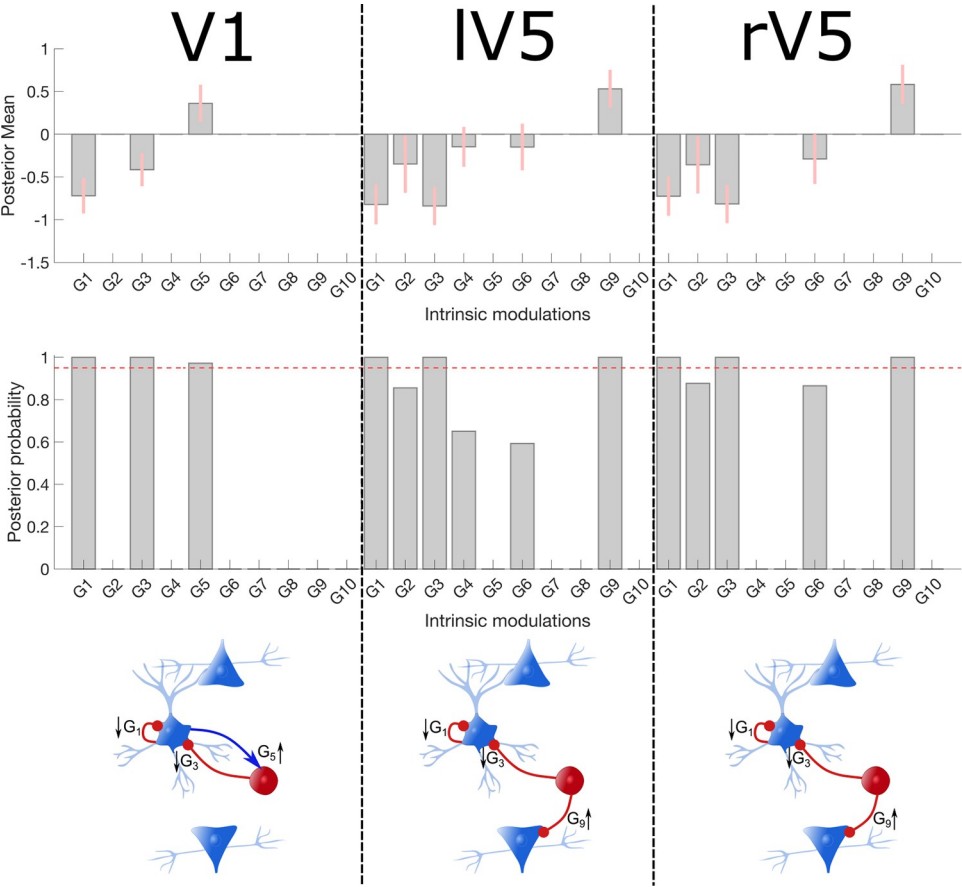

**Fig 5. Intrinsic connectivity modulations.** The figure reports the group-level results of the intrinsic connectivity modulations associated with EC states (relative to EO) in the three sources of interest. More specifically, the mean posterior and 90% Bayesian confidence interval (pink error bars) after the greedy search algorithm and Bayesian model averaging are shown in the top row. The middle row shows the corresponding posterior probabilities of the intrinsic modulations, the pink dotted lines correspond to a Pp of .95. Finally, the bottom is a schematic presentation of the modulation with Pp>.95 and the direction of the effect. Inhibitory connections are shown in red and excitatory in blue.

modulations. Interestingly, our results showed that the intrinsic connections in V5 play a relatively larger role compared to the extrinsic connections and V1 intrinsic connections. Most inhibitory connections to SS-cells—the target population for endogenous neuronal fluctuations—decreased during EC. Overall, we found that decreased inhibitory connections within the higher order visual cortex seem to play an important role in underwriting EO-EC alpha power differences.

Given the role of inhibitory intrinsic connections found here, it is worth noting that cortical inhibition is largely mediated by GABAergic connections, while excitation is mediated by glutamatergic connections [51]. In a recent review paper by [52], the author discusses studies that used pharmacological modulation to study physiological mechanism underlying alpha rhythms. In their review, several findings are discussed in light of the so-called alpha power as inhibition principle [53]. Briefly, this principle states that alpha oscillations serve a functional inhibitory role which is implemented through physiological inhibition (generated by GABAergic interneurons). Following this principle, one would expect increased alpha in case of increased physiological inhibition. However, [52] reviewed several lines of evidence showing pharmacologically enhanced inhibition results in decreased rather than increased alpha. In

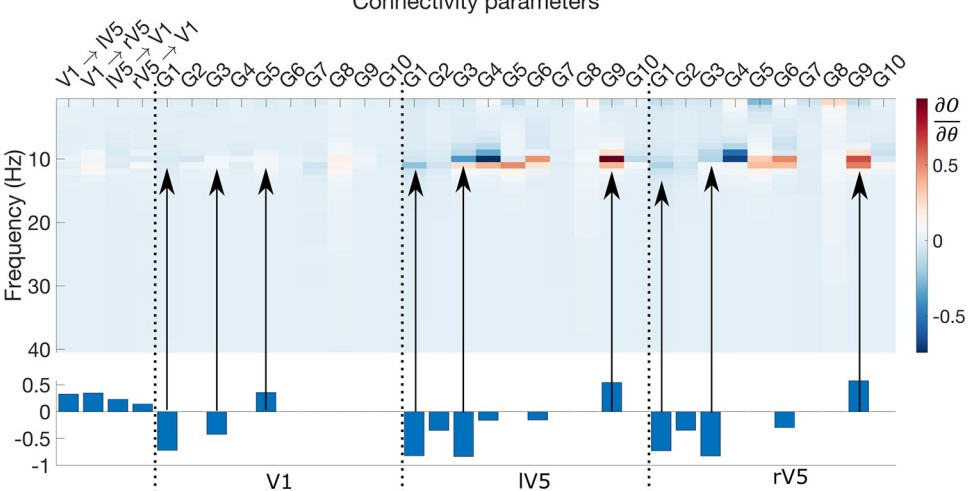

**Fig 6. Sensitivity analysis.** The results of the sensitivity analysis—averaged across subjects—are shown in image format. The group BMA results (i.e. the middle panels of **Figs 4 and 5**) are provided in the lower part for comparison. Simply put, positive (red) values indicate that an increase of a parameter at the posterior mean would result in increased power at that specific frequency bin. A negative value (in blue) means an increase of a particular parameter results in a decreased power. The arrows indicate the sensitivities for the significant intrinsic connectivity modulations that are shown in the lower part of the figure. $\frac{\partial O}{\partial \theta}$ is the Jacobian of the power spectrum of the generative model with respect to the model parameters of interest (intrinsic and extrinsic) which are shown for all modelled frequency bins.

addition, some studies have found that sub-anaesthetic doses of ketamine (i.e. a glutamatergic excitatory NMDA receptor blocker) resulted in decrease posterior alpha power in resting-state [54, 55]. According to [56], inhibition plays an important role in rhythmogenesis, either in an interneural network or via excitatory-inhibitory loops. In sum, these studies are in line with our findings regarding the importance of local inhibition in the generation of alpha rhythms during rest.

The clinical relevance of comparing eyes-open vs. eyes-closed data using DCM can be appreciated knowing that the (normalized) difference between EC and EO (which is some-times termed alpha-reactivity) is related to specific neurological conditions [31, 57, 58]. However, reduced alpha reactivity can manifest itself in different ways. The reduction in reactivity can be due to increased EO alpha power and unaltered EC power, or decreased EC power and unaltered EO power. For example, [57] showed that in Alzheimer disease patients, EC alpha was reduced while EO alpha was not compared to healthy controls. The reverse was true for Lewy body dementia patients. Using DCM, one could identify which parameter of the model employed here could explain these observed 'alpha reactivity' differences. These parameters could possibly be useful as an aiding tool for differential diagnosis."

Several studies using biologically inspired models, fitted to EEG data, have been conducted in the context of EO-EC alpha power differences. In two recent studies by [59] and [60] the authors used a neural mass model of the same data set used in our study. In the first study, the authors investigated parameter identifiability of a 22-parameter neural mass model based on the EC data alone. They found that, using sampling-based inversion scheme a single parameter controlling inhibitory synaptic activity is directly identifiable. In a follow up paper, the authors extended the model by incorporating modulatory parameters used for explaining EO-EC power differences. Their main finding was that a single modulatory parameter seems to explain best the alpha power difference; namely, a parameter controlling the tonic excitatory input to inhibitory populations. The authors argue in light of previous findings, that this external input is likely to be of thalamic origin. In relation to our modelling approach, several

differences are important to consider. First, we used a neural mass model of multiple spatially defined and coupled occipital sources. This is to be contrasted with earlier studies, where no reference to coupled regions was made. Second, DCM combines a neural model of how different subpopulations within and between cortical sources interact, with a forward model of how post-synaptic potentials are mapped to observed data (here channel cross spectral densities). In comparison, the earlier studies mentioned above did not include an observation model. Thus, neural activity was not decoupled from volume conduction and observation noise (channel noise). Third, our approach used a variational Bayesian inversion scheme, which provides a lower bound on the log-model evidence, necessary for Bayesian model comparison. In other words, we identified the most plausible model, where both model fit and complexity were considered in scoring alternative models. In [59] used particle swarm optimization and constrained half of the model parameters to be the same between the EO and EC conditions. They found that only 1 modulatory parameter provided the best explanation for generating EC-EO alpha differences. On the other hand, here we found that several modulatory parameters were identified for explaining spectral differences between EO and EC. Using BMC, we showed that both intrinsic and extrinsic connectivity parameters are necessary to explain the data. Fourth, in [60] and [59], the authors only used data from Cz to estimate the parameters of the model, while in the current work we used data from all EEG channels (projected to a reduced space).

In another related study, using empirical EO-EC EEG data for estimating the parameters of a neurophysiological model, the authors found multiple parameters that explained the difference between EO and EC [61]. Similarly, to the model by [59] these authors used data from a single electrode and did not include an observation model. They considered a thalamo-cortical model including intracortical and thalamocortical pathways and four type of neurons: cortical pyramidal (excitatory) and inhibitory neurons, thalamic reticular and thalamo-cortical relay neurons. They found that strong positive (excitatory) cortico-thalamic feedback and longer time constants underlie EC alpha power. One of the major strengths of this study is the incorporation of thalamocortical interactions, which is lacking in the current study. In principle, it is possible to incorporate the thalamus as a hidden source in DCM (i.e., the states of the hidden node do not contribute directly to the observed responses) to investigate bidirectional effect of thalamo-cortical dynamics [62]. In essence, this approach would be the same as extending the current neural mass model to include additional subpopulations representing the thalamus. This approach was undertaken in the recent model by [10] by including excitatory and inhibitory neural populations in the thalamus. Their temporal dynamics are given by the well-known model of [63] that describes thalamic oscillations [64, 65]. The model includes thalamocortical relay (TC) and thalamic reticular nucleus (TRN) neurons. TC neurons project to the cortex, while TRN neurons surround the thalamus and regulate TC neuron activity by sending inhibitory signals. This model could be used in DCM to explain multimodal data (EEG and fMRI) from the thalamus and reveal differences in laminar dynamics [40, 41]. Alternatively, it would be interesting to apply DCM to intracranial recordings where both thalamic and cortical areas are recorded simultaneously. Using DCM, we built a parsimonious model that can accommodate alpha power differences generated via changes in either within or between higher and lower order visual areas. Following [8], that found alpha traveling waves from higher to lower visual areas we considered three visual sources V1, left V5 and right V5. It is possible that other visual areas such as V2 and V4 as well the thalamus contributes to alpha rhythm generation. In our model, these contributions are modelled as spectral innovations with white and pink components; whose amplitude and slope are estimated for each brain area separately. Alternatively, one could implement Bayesian model comparison to identify the functional network, similarly to [66].

Functionally, two different brain configurations have been associated with EO and EC resting-state condition: an exteroceptive state associated with attention, vigilance and ocular motor activity and an interoceptive state associated with mental imagery and multisensory activity [67–69]. Considering our results, one could suggest that inhibition in higher order visual areas are the local manifestations of an interoceptive state that is triggered by eye closure.

Differences in power in a given band could be explained by changes in slope/aperiodic part apart from modulations in pure oscillations [70]. In DCM, the shape of the observed spectra is determined by the parametrized 1/f neural fluctuations (a.k.a., innovations) and importantly the transfer functions that govern 'spectral bumps' in the output [17, 45]. In this study, we assumed that during both EC and EO, the spectral shape of the innovations remains the same and that differences are due to changes in connectivity. A possible avenue for future research concerns the shape of the neural innovations driving V1 and V5. The current model could be augmented by allowing condition specific changes in either the height, slope or both of the 1/f form of the neural innovations. In addition, condition specific changes in the synaptic time-constants of the different neural populations could be examined.

Considering the aforementioned evidence, we suggest that dynamic causal modelling of resting-state EO and EC conditions might provide a mechanistic insight into intrinsic physiological mechanisms. This could be relevant for quantitative insights in clinical studies but also studies that use pharmacologically altered states of consciousness.

## Supporting information

**S1 Fig. Spectral densities of the removed subjects and four example subjects.** This figure displays the spectral data of the 4 data modes of the 4 removed subjects (top 4 rows) and 4 randomly selected subjects (bottom 4 rows). The 4 columns are the 4 data modes. Power spectra are shown for the EO (in red) and eyes closed (in blue) condition. The spectral densities were obtained by means of a Bayesian autoregressive model (see main text).
(PNG)

**S1 Appendix. Variational Bayesian model inversion.** This appendix summarizes the variational Bayesian framework used in this paper.
(DOCX)

**S2 Appendix. Notes on log-scaling prior.** These notes provide more details on the priors of the log-scaling parameter.
(DOCX)

## Author Contributions

**Conceptualization:** Frederik Van de Steen, Daniele Marinazzo.

**Formal analysis:** Frederik Van de Steen.

**Funding acquisition:** Daniele Marinazzo.

**Investigation:** Frederik Van de Steen.

**Methodology:** Frederik Van de Steen.

**Visualization:** Frederik Van de Steen.

**Writing – original draft:** Frederik Van de Steen.

**Writing – review & editing:** Frederik Van de Steen, Dimitris Pinotsis, Wouter Devos, Nigel Colenbier, Iege Bassez, Karl Friston, Daniele Marinazzo.

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
