## [Decision Letter · Decision Letter 0]

6 Jul 2022

Dear Dr. Van de Steen,

Thank you very much for submitting your manuscript "Dynamic causal modelling shows a prominent role of local inhibition in alpha power modulation in higher visual cortex" for consideration at PLOS Computational Biology. As with all papers reviewed by the journal, your manuscript was reviewed by members of the editorial board and by several independent reviewers. The reviewers appreciated the attention to an important topic. Based on the reviews, we are likely to accept this manuscript for publication, providing that you modify the manuscript according to the review recommendations.

Three reviewers have looked at your manuscript and are generally enthusiastic about it. However, they all have questions about the experiments, methods and results. I therefore suggest you submit a revised version of the manuscript in which you carefully address these issues.

Sincerely,

Marieke Karlijn van Vugt, PhD

Associate Editor

PLOS Computational Biology

Lyle Graham

Deputy Editor

PLOS Computational Biology

[LINK]

Three reviewers have looked at your manuscript and are generally enthusiastic about it. However, they all have questions about the experiments, methods and results. I therefore suggest you submit a revised version of the manuscript in which you carefully address these issues.

Reviewer's Responses to Questions

**Comments to the Authors:**

Reviewer #1: This paper seeks to identify the physiological origin of increased alpha activity which occurs in (most) individuals upon closing of the eyes. Dynamic causal modelling is used to fit publicly available EEG data from over 100 subjects. The model is composed of microcircuits in the V1, right and left V5 regions, each of which consists of spiny stellate, superficial pyramidal, deep pyramidal, and inhibitory subpopulations. The hypothesis is that changes in alpha activity are due to changes in the connections between neuron subpopulations. By inferring the presence of these connections, specifically the intrinsic connections (within a region) and the extrinsic connections (between regions), and how the connection strengths are modulated when going from eyes-open to eyes-closed conditions, the authors conclude that much of the increase in alpha activity upon closing of the eyes can be attributed to the reduction in intrinsic inhibition activity between certain subpopulations.

Modulation of alpha activity upon opening or closing of the eyes is one of the most ubiquitous and repeatable features of the EEG. The conclusions presented here are significant for this reason alone and are potentially relevant to clinical and pharmocological applications.

The previous literature on the subject is described well and a detailed comparison is presented with the recent study published in this journal (Ref [43]), which also concluded that cortical inhibition was important in modulating alpha activity. An important advance is the use here of cross spectral densities. Can the authors comment on whether any of the parameters could have been estimated by modelling the power spectrum alone?

In general I found the work convincing although I did have to work through several of the references to get there.

I still have a few questions having to do with the data analysis.

(1) p.9 Line 203. “The first 4 eigenmodes of the prior data covariance are used to project the channel data into a reduced sensor space”. Does this mean that only data which is consistent with the model is being used in the fit? Approximately what proportion of the variance is discarded? I was not able to find much indication of this in Ref [40].

(2) p.10. Table 1. I could not find where the coefficients specified for the different parameters (in the column “Parameterisation”) was explained. For example, where do the values [4,4,8,4,4,2,4,4,2,1] come from? How well known are they, and how might different values influence the conclusions? e.g. G3 has the largest coefficient and also has the largest (absolute) posterior mean value (Figure 6, top).

(3) p.11. Line 218. “In total, 5 subjects were removed from the group analysis.” This represents ~5% of the subjects. Were there particular features of these discarded datasets that made the model fit inadequate?

I also suggest some minor corrections or clarifications:

p.7 Figure 1. “Intrinsic” is misspelled twice, next to the figure.

p.10. Table 1. The reference in the caption is misspecified.

p.10. Line 213. Is the ‘IMG’-option that specifies the forward model an option in the code?

p.13/14. Figure 3 is shown twice, once on each page.

p.16. Figure 5 is not referred to in the body of the text.

p.17. Figure 6. The colour bar is labelled “dO/d\\theta”. I could not find where “O” is defined.

p.17. Line 310. “Technically, we are numerically evaluating the Jacobian of the generative model of the extrinsic and intrinsic connectivity modulations at their posterior means.” I assume the output being evaluated is the power in a particular frequency channel? Is this “O” in Figure 6?

p.17. Line 315. “Positive (red) and negative (blue) values indicate that increasing or decreasing the parameter would result in increased and decreased power, respectively.” The use of “respectively” is a bit confusing here. Can you separate this into 2 sentences?

p.18. Line 342. “In two recent studies [43] and [43]”. Presumably one of these references should be [44]?

-Damien Hicks

Reviewer #2: The study aims to understand the difference between eyes open and eyes closed brain states by inverting models of EEG. Unfortunately the manuscript lacks clarity and accuracy in presentation in a number of places, which makes assessing it difficult (see some details below). Also, part of the study using sensitivity analysis is limited to a single perturbation. More analysis is required there in order to support any conclusions.

clarity in introduction / motivation

- In the introduction it is stated that cortico-cortical interactions play more of a role than thalamo-cortical interactions. Could the authors please expand on the evidence for prominence of cortical interactions above thalamocortical interactions.

- in the motivation for dynamic causal modelling it is stated that measures like coherence "do not provide a unique or complete description of the underlying data generating process...". It is unclear what a complete description would be, what you mean by data generating processes, and in what sense dynamic causal modelling would provide a complete description of this. It would be good to clarify this, or give clearer motivation for the use of dynamic causal modelling.

- rather than specifically addressing the challenges laid out, isn't it rather the case that DCM would give complementary information?

- the authors should expand on in what sense DCM has been validated - this seems like a strong statement that potentially needs clarifying / caveating.

- the statement that DCM yields efficient source reconstruction also should be expanded. Is that true for scalp EEG recordings of alpha rhythm, for example, is it true in general, or is this specific to the setup in reference 18?

- "DCM .. provides a solution to the inverse problem.." Which specific inverse problem are you referring to?

- ".. where both amplitude and phase information are used..." are you referring to phase information of a single signal, between signals or both?

- rather than only stating that the use of 3 distinct sources in their model is inspired by reference 8, the authors should give detailed reasons and evidence for why they made this assumption.

clarity in methods:

- In the methods section and at the start of results there is reference to the permutation test that was carried out. For clarity, please make the description of this more explicit, and also include the motivation for doing this analysis. What t-test was performed exactly, i.e. distributions of which data were used? It is only stated that the EO and EC conditions were swapped.

- The model needs to be described much more accurately and please be consistent with notation. Some examples: use of open circle and asterisk to denote multiplication (presumably) in different places; sigma only mentioned in the figure legend, and without giving its mathematical definition; no explanation of the biophysical meaning of the equations given; observation model not explicitly given

- The inference methods should be more accurately described, for example: no definition of any of the thetas is given; there appears to be inconsistency in subscript between thetas ("Y" or "G", what do these mean?); no motivation is given for the parameter ranges; for readers of plos CB at least some details of the inference scheme should be given.

clarity in results:

- figure 2: it might be old fashioned, but I find referring to components of figures in order from left to right helpful. labels a, b etc can also help. Consistency is important - we have channels numbered or given their letter/number labels. Please alter so we can compare the two components of figure 2. Are there units for power?

- figure 3 is actually repeated twice in my pdf. Please explain more explicitly what you mean by taking all combinations of parameters.... What is the Null model?

- Models without V5 have lower evidence - is this because there are more parameters in v5 as there are two of them? Please provide the formula for model evidence.

- Please provide some exemplar fits so that we can see visually what the tradeoff between complexity and match to data is (presuming that is in the measure of evidence).

- Figure 4: how does the parameter estimate relate to any of the Gs or thetas etc? Please be explicit. What is the posterior probability? Probability of what exactly?

- rV5->V1 has uncertainty encompassing zero, so is it true to say that all extrinsic connections increase?

- it seems figure 5 is not discussed in the text at all

- why is e^-6 chosen as an increment? How does this result change if different values are chosen? What are the posterior parameter values that are fixed? Is there a simple explanation for the changes being largest in certain parameters in figure 6, such as the perturbation made to them is larger on the scale of their values? What if you make proportional changes? Please provide axis labels. What is dO/dtheta? "changes in intrinsic connectivity have a larger effect.." this is true for V5 but not for V1.

Discussion

- line 342: reference 43 repeated.

Reviewer #3: This was a well described study using canonical microcircuit modelling and DCM to investigate intrinsic and extrinsic connectivity changes that may underlie the differences in alpha oscillations between eyes open and closed resting state. The study is well thought out and illustrated. My comments are generally minor.

I must note also that there are numerous formatting errors that while they won’t influence the final product after typesetting and editing, it made the paper unnecessarily difficult to read/review. This included numerous errors with in text referencing for tables and figures. Figure placement was problematic (sometimes half way through its own caption) see Results point 2 and 3. There was inconsistent referencing styles (number format and author date) as well as references to the same paper twice where there should have been two separate references.

Methods

1. Why do you run a 60 Hz filter if you then go on to high pass filter the data at 45 Hz?

2. Where you describe using the model the ‘canonical microcircuitry’, it would be useful to describe exactly which model you used. If you used the standard CMC out of the spm toolbox please describe the function you used that determines the model, and the SPM toolbox version. Then clearly refer to the modifications and relevant papers – which already do emerge in the following section. It would be ideal to be able to clearly evaluate this while reading to both smooth comparisons between DCM papers and any replication attempt.

(I did note afterward the code is available on GitHub and the implementation is clearly described in the ReadMe - so referencing that this is the case in text would also help).

3. For Figure 1, in the caption it would be useful to refer to Table 1 for complete interpretation of the equation notation, or describe/link this all up more clearly in the following section, or clearly reference the relevant papers.

4. Fix referencing error in Table 1: “Table 1. Parameters of neural model (see Error! Reference source not found. for illustration of the neural model)”

Results

Results were clearly presented and well furnished with figures – really just a couple of notes that impeded reviewing and ought to be fixed to help any further revisions:

1. Duplicate Figure 3 appears in text in section “PEB model selection and parameter averaging”. This appears to be related to an error in the way automatic figure referencing is happening in text.

2. Figure 6 appears halfway through your figure caption

3. Often the figures appear to randomly be placed mid paragraph, sometimes before their first reference in text.

Discussion

1. Duplicate references means one is probably missing? Line 342: “In two recent studies by [43] and [43]”

2. Optional, but is it worth commenting on the implication of comparing EC vs EO data between clinical/pharmaco studies based on the mechanistic differences described here?

3. Also just a suggestion, but given the paragraph starting “Given the role of inhibitory intrinsic connections found here,…” directly describes the study data and its impact/implications should it be the second paragraph of the discussion? At the moment it’s rather buried.

This section may be of future use to clinical and pharmaco resting studies. It will also be the section where the people who don’t understand/use DCM and canonical microcircuits will learn how to interpret the model findings in neurobiological terms.

4. What do you mean by innovations in the paragraph on 1/f fluctuations? Do you mean innervation?

**Have the authors made all data and (if applicable) computational code underlying the findings in their manuscript fully available?**

Reviewer #1: Yes

Reviewer #2: Yes

Reviewer #3: Yes

PLOS authors have the option to publish the peer review history of their article (what does this mean?). If published, this will include your full peer review and any attached files.

Reviewer #1: **Yes: **Damien Hicks

Reviewer #2: No

Reviewer #3: No

Figure Files:

Data Requirements:

Reproducibility:

References:

---

## [Decision Letter · Decision Letter 1]

5 Dec 2022

Dear Dr. Van de Steen,

Thank you very much for submitting your manuscript "Dynamic causal modelling shows a prominent role of local inhibition in alpha power modulation in higher visual cortex" for consideration at PLOS Computational Biology. As with all papers reviewed by the journal, your manuscript was reviewed by members of the editorial board and by several independent reviewers. The reviewers appreciated the attention to an important topic. Based on the reviews, we are likely to accept this manuscript for publication, providing that you modify the manuscript according to the review recommendations.

Many thanks for your revision. The reviewers are mostly happy with the manuscript, but point out some typos. I think it would be worth fixing those before publication, which is why I recommend a minor revision. Congratulations!

Sincerely,

Marieke Karlijn van Vugt, PhD

Section Editor

PLOS Computational Biology

Lyle Graham

Section Editor

PLOS Computational Biology

Many thanks for your revision. The reviewers are mostly happy with the manuscript, but point out some typos. I think it would be worth fixing those before publication, which is why I recommend a minor revision. Congratulations!

Reviewer's Responses to Questions

**Comments to the Authors:**

Reviewer #2: Many thanks for the thorough revision. The manuscript is much more accessible now and I think it is suitable for publication. There are just two minor points that i think should be addressed and a few typos that i noticed:

1. line 227- page 10. Can you give a figure in the manuscript to explain / quantify what "relatively uninformative" prior refers to? Perhaps you could show a plot of the prior distributions in supplemental so the reader can assess this for themselves?

2. line 389 page 16. It is stated that the largest negative sensitivities are for G3 and G1. It is qualified that you are looking at the significant intrinsic modulation, but it is worth pointing out and explaining that the largest sensitivities are actually for G4. It's interesting that this parameter can cause large deviations in the spectrum but has not been identified as an important parameter in the DCM.

typos:

page 7 - should the "T" in T-value and T-test be lower case?

bottom of page 9, line 223, "where chosen" -> "were chosen"

line 357, pg 15: "observed a significant reduced" -> "observed a significant reduction"

Reviewer #3: The authors have addressed my concerns. Thank you.

**Have the authors made all data and (if applicable) computational code underlying the findings in their manuscript fully available?**

Reviewer #2: Yes

Reviewer #3: Yes

PLOS authors have the option to publish the peer review history of their article (what does this mean?). If published, this will include your full peer review and any attached files.

Reviewer #2: No

Reviewer #3: No

Figure Files:

Data Requirements:

Reproducibility:

References:

---

## [Editor Report · Decision Letter 2]

16 Dec 2022

Dear Dr. Van de Steen,

We are pleased to inform you that your manuscript 'Dynamic causal modelling shows a prominent role of local inhibition in alpha power modulation in higher visual cortex' has been provisionally accepted for publication in PLOS Computational Biology.

Best regards,

Marieke Karlijn van Vugt, PhD

Section Editor

PLOS Computational Biology

Lyle Graham

Section Editor

PLOS Computational Biology

Thanks so much for making the final revisions. The paper is now ready for acceptance. Congratulations!

---

## [Editor Report · Acceptance letter]

22 Dec 2022

PCOMPBIOL-D-22-00324R2 

Dynamic causal modelling shows a prominent role of local inhibition in alpha power modulation in higher visual cortex

Dear Dr Van de Steen,

I am pleased to inform you that your manuscript has been formally accepted for publication in PLOS Computational Biology. Your manuscript is now with our production department and you will be notified of the publication date in due course.

With kind regards,

Zsofia Freund
